# Postural stability of 5-year-old girls and boys with different body heights

**Magdalena Plandowska**  *, **Małgorzata Lichota, Krystyna Górniak**

Faculty of Physical Education and Health, Jozef Pilsudski University of Physical Education, Warsaw, Poland

* magdalena.plandowska@awf-bp.edu.pl

## Abstract

### Background

Postural stability is one of the determinants of proper body posture and a condition for developing motor abilities in every human being. The measurement of the centre of pressure (COP) location and displacement is the most common technique of postural stability assessment.

### Objective

The aim of this study was to assess differences in postural stability depending on sex of 5-year-old children with different body heights.

### Methods

A study included 435 healthy children (200 girls and 235 boys) born in 2010 whose parents gave a written consent to their participation in the project. Postural stability was assessed with the use of the dynamographic platform (Zebris FDM 1.8). The assessment of postural stability was based on COP shift parameters (sway path length of COP and average velocity of COP) and COP surface area parameters (area of the ellipse, length of ellipse in the anterior-posterior direction and length of the ellipse in the medial-lateral direction). Body height was measured with Holtein anthropometer and the obtained values were compared with percentile ranks determined by the WHO.

### Results

The analysis of the parameters describing postural stability in the examined children revealed dimorphism. For the COP shift parameters and COP surface area parameters, the level of statistical significance was recorded for girls and boys. Girls achieved lower results of these parameters than boys regardless of their body height. In the groups of normal- and tall-statured children, differences between the genders were statistically significant.

### Conclusions

The present study characterised sex differences in postural stability of 5-year-old children. Sex-related differences were found during a natural stance for all COP parameters. Girls

**Data Availability Statement:** All relevant data are within the manuscript.

**Funding:** The authors received no specific funding for this work.

**Competing interests:** The authors have declared that no competing interests exist.

**Abbreviations:** AoE, area of the ellipse; COP, centre of pressure; LoEAP, length of the ellipse in the anterior-posterior direction; LoEML, length of the ellipse in the medial-lateral direction; SD, standard deviation; SP, sway path length of COP; V, average velocity of COP.

maintained a two-legged standing position with lower sway velocity and a smaller range of sway than their male counterparts. Normal- and tall-statured girls demonstrated better postural stability significantly more often than boys.

## Introduction

Being able to maintain standing balance is the basis of a child's motor development. The fifth and the sixth year of life is a period of positive changes in physical fitness and motor coordination. During childhood, children begin to learn and acquire many fundamental motor skills. Adequate acquisition and command of fundamental motor skills at the end of preschool age (around the age of six) have been considered as crucial elements in the development of specialised and more complex motor skills [1, 2]. Proper postural stability is the basic condition when it comes to improving motor skills specific to children [3].

Vertical positioning of the body axis in relation to the support area is a characteristic feature of body posture. To achieve and maintain an upright standing position, efficient postural control is required. Postural control is a complex process that depends on the integration between the sensory and musculoskeletal systems [3, 4]. Upright posture control includes minimising body sway so that the vertical projection of the centre of mass (COM) is maintained within limits of the base of support. Body sway is treated as an indicator of the balance system efficiency. It can be assessed by measuring the deviations in the location of the vertical ground reaction force vector (the centre of pressure, COP) on the supporting surface. In the sagittal plane (AP–anterior-posterior), these forces are generated by muscles responsible for ankle joint movements, while in the frontal plane (ML–medial-lateral), forces are produced by hip abductors and adductors [5, 6, 7]. Larger COP oscillations are considered to be the signs of postural instability.

Body posture and stability change in the course of ontogenetic development. Movement patterns, muscle tone and the sensory system also develop in the process of ontogenesis. Although somatic changes in a child may affect the level of body posture control, it is the development of the sensory nervous system and vestibular system that affect body posture stability to a larger extent [8, 9, 10]. The control of body balance in children becomes more and more efficient as the latent period of muscle contraction shortens and the velocity of neuromuscular transmission increases [11]. The quality of the transmission of sensory and motor stimuli is also affected by the myelination of nerve fibres. Muscle strength increases together with the development of muscle tissue, innervation, proprioception and an ability to engage a maximal amount of neuromuscular motor units in a movement. In a child between 4 and 7 years of age, this increase constitutes 75% of the baseline strength from the age of 4. Moreover, around the age of 5, the process of the development of extensor mass accelerates and extensor tonus balances the tension of flexors. This process facilitates the maintenance of an upright body position. In a child aged 5–7, the mass of extensors is 2–3 times bigger than the mass of flexors in particular lower limb joints [12]. By 6 years of age, though, children have achieved adult levels in some specific aspects of postural control [9, 13, 14]. The biggest developmental changes regarding balance control occur in children aged 6–8 [15, 16, 17].

Research on postural stability performed with the use of various platforms is carried out more and more often and concerns different age groups. However, these studies are usually of comparative character or reflect postural stability of children and youth with certain developmental dysfunctions. Researchers often carry out studies on children of both genders together

or divide children into two age categories by combining 3- and 4-year-olds in one group and 5- and 6-year-olds in another. In turn, publications on postural stability of small healthy children born in the same year are less common.

In the 5[th] year of life, gender has proved to be an important factor in acquiring and mastering fundamental motor skills. Sex differences exist during childhood and may be credited by various factors, i.e. biological, physical, psychological. Maturational differences between girls and boys do not only exist regarding physical, hormonal, and sexual development, but also in terms of central nervous structures. It was indicated that brain structure and development differ between sexes during infancy [18, 19, 20]. Girls are better at skills which require balance and rhythm as well as precision, e.g. tiptoeing, walking on a balance beam or walking foot by foot, one-leg balance on the preferred leg, which are stimulated by the vestibular system. In turn, boys achieve better results in activities which require more speed and strength [21, 22, 23]. Webster et al [24] also showed that boys engaged more in total physical activity, moderate-vigorous physical activity, and less in sedentary behavior.

Can we, therefore, expect postural stability of girls to be better than in boys? The aim of this study was to assess differences in postural stability depending on sex of 5-year-old children with different body heights. It is hypothesised that girls have better postural stability than boys, reflected by a decrease in the amount of sway among girls.

## Materials and methods

### Participants

The study was carried out on 435 healthy 5-year-olds (62.1±3.47 months) from Poland including girls (n = 200) and boys (n = 235), which constituted 76% of the total number of 5-year-olds attending kindergartens in Biala Podlaska in the school year 2015/2016. The inclusion criteria were as follows: a) being born in 2010, b) attending a kindergarten in the school year 2015/2016, c) participants of both sexes, d) good general health. Children were excluded from the study if they: a) had any neurological and musculoskeletal disorders confirmed by their parents/guardians, b) had bad general health. No child had a dysfunction that could impair their ability to maintain balance.

Written informed consent was obtained from the parents/guardians of all the children enrolled in the study. The research was conducted within the statutory research project DS. 246 titled "Psycho-physical development of 5-year-olds from Biała Podlaska", which was accepted by the Research Commission of the Faculty of Physical Education and Sport in Biała Podlaska and the Senate Scientific Research Ethics Commission of Józef Piłsudski University of Physical Education in Warsaw (SKE 01-01/2014). The examinations on postural stability were performed between November 2015 and January 2016 in the Body Posture Laboratory at the Regional Centre for Research and Development of the University College in Biała Podlaska. All the examinations were made in the morning hours in the laboratory rooms in which safety, intimacy and basic hygienic requirements were ensured.

### Test procedure and protocol

**Stage I–preparation.** At the preparation stage (April–August 2015), talks with kindergarten directors and teachers were initiated and information meetings with parents were held in order to obtain their consent to include their children in the project. The purpose and procedure of this study were explained in detail to parents and kindergarten teachers.

**Stage II–initial medical check-ups.** Prior to the examinations, the children underwent initial medical check-ups which qualified them to participate in the study. General health state

was assessed on the basis of heart and lungs auscultation in a standing position and the examination of blood pressure using the auscultation method in a sitting position [25].

## Stance analysis

Postural stability was assessed with the use of the dynamographic platform Zebris FDM 1.8 (Force Distribution Measurement, 208x56 cm, 120 Hz, System Stance Analysis, Medical GmbH, Germany), which measures the COP signal. The platform was connected to the WinFDM software for analysing body sway. The measuring device was calibrated prior to the examination of every child. During the measurement, the COP signal was registered. After the registration, the system automatically performed basic analyses of the registered signals.

Children were examined individually while standing on a platform in a place indicated by the researcher. The task involved maintaining a two-legged stance with eyes open for 30 seconds. Children were standing barefoot with their legs straightened in the knee joints and their feet parallel to each other, their upper limbs along their torsos and their heads in the Frankfurt plane.

In case of noticeable movements of the head, upper limbs or lower limbs, a child repeated the test a few minutes later. The duration of a single test made it possible to detect COP oscillations reflecting real conditions and at the same time not evoking a negative attitude of children [26].

The assessment of postural stability was based on five sway parameters: COP shift parameters (sway path length of COP and average velocity of COP) and COP surface area parameters (area of the ellipse, length of the ellipse in the anterior-posterior direction and length of the ellipse in the medial-lateral direction):

- sway path length of COP (SP) [mm]–defines the total length of path marked by the COP; the sum of distances between the locations of the COP constitutes the path length;

- average velocity of COP (V) [mm/s]–defines mean velocity at which the COP moves; this parameter indicates the speed of changes in the COP location, which reflects the speed of postural reactions while standing;

- area of the ellipse (AoE) [mm$^2$]–defines the size of the area marked by the COP; ellipse area includes 95% of the COP measurement points; this parameter makes it possible to assess the size of the area of the COP movement on the support surface;

- length of the ellipse in the anterior-posterior direction (LoEAP) [mm];

- length of the ellipse in the medial-lateral direction (LoEML) [mm].

Each sway parameter was evaluated as follows: the higher the parameter score, the greater the sway and the worse the postural stability.

## Body height measurement

The characteristics of postural stability of the examined children required assessing sex-related differences in the groups of children at a similar level of physical development taking into account body height.

The measurement of body height is a standard manner of monitoring a child's growth and development [27]. Body height was measured with Holtein anthropometer with the accuracy of 0.1 cm. The examined children born in 2010 manifested various values of body height increase. Mean values of this parameter in girls were 1.14±4.89 and in boys– 1.15±5.38, and differences between them were statistically significant (U = 26664, p = 0.02).

In order to minimise differences in body height, body height values of study participants were compared to the body height values created by WHO for children in particular months of life. WHO 2006 standard is the outcome of a multinational study. Growth charts are widely used as a clinical tool to monitor growth in individual children. Height status was classified according to age and gender into low stature (-1) (<25th percentile), normal stature (0) (≥25th percentile to ≤75th percentile) and tall stature (1) (>75th percentile) based on the percentile ranks determined by the WHO [28]. In this way, children at a similar level of physical development were selected.

## Statistical analysis

The collected material was organised and analysed with the use of Statistica 13 calculation software by Statsoft (PL). The parameters were described using basic measurements of descriptive statistics, i.e. mean, standard deviation (SD), median (Me). The compliance of the results with normal distribution was checked with the Shapiro-Wilk test. Data normality was rejected, so the Mann-Whitney U test was used to examine sex differences regarding body height values and postural stability parameters. Statistical significance was set at $p < 0.05$.

## Results

The results of measures describing COP shifts recorded with regard to sex are presented in S1 Fig. For the sway path length (SP) and average velocity (V) of the COP, the level of statistical significance was recorded for girls and boys. The analysis of the COP shift parameters describing the process of maintaining static balance in the examined children revealed that girls achieved significantly lower values of these parameters than boys (sex effect for SP: U = 17312.5, p<0.001; gender effect for V: U = 17229.5, p<0.001). Moreover, sex differences in COP shift parameters taking into account body height categories were noted. Girls achieved lower results of these parameters than boys regardless of their body height. In the groups of normal- and tall-statured children, differences between the sexes were statistically significant (normal-statured, sex effect for SP: U = 6655.5, p = 0.002, sex effect for V: U = 6608, p = 0.001; tall-statured, sex effect for SP: U = 2152, p<0.001, sex effect for V: U = 2145.5, p<0.001). The percentage of children taking into account body height categories determined by the WHO is shown in Table 1. Mean and median values of measures describing COP shifts recorded in girls and boys taking into account body height categories determined by the WHO are presented in Table 2.

A statistically significant sex effect was observed for COP surface area parameters (S1 Fig). Girls achieved significantly lower values of these parameters than boys (sex effect for AoE: U = 18044.5, p<0.001; sex effect for LoEAP: U = 20418.5, p = 0.018; sex effect for LoEML: U = 18631, p<0.001). The analysis revealed sex differences for COP surface area parameters in

**Table 1. The percentage of children taking into account body height categories determined by the WHO.**

|  |  | Body height categories determined by the WHO n (%) | | |
|---|---|---|---|---|
|  |  | **1** | **0** | **-1** |
| Sex | Girls (n = 200) | 69 (34.5) | 127 (63.5) | 4 (2.0) |
|  | Boys (n = 235) | 94 (40.0) | 135 (57.4) | 6 (2.6) |
| All (n = 435) | | 163 (37.5) | 262 (60.2) | 10 (2.3) |

Note: n—number of children; %—percentage; (1) tall-statured children; (0) normal-statured children; (-1) low-statured children.

**Table 2. Mean (SD) and median values of COP parameters taking into account sex and body height categories determined by the WHO, and significance levels from the Mann-Whitney U test.**

| Body height categories determined by the WHO | COP parameters | Girls | | Boys | | p value |
|---|---|---|---|---|---|---|
| | | Mean±SD | Me | Mean±SD | Me | |
| 1 (n = 163) | SP [mm] | 517.7 ±171.33 | 503.0 | 621.2±186.60 | 602.4 | <0.001 |
| | V [mm/s] | 18.0±5.90 | 17.6 | 21.6±6.50 | 20.8 | <0.001 |
| | AoE [mm²] | 792.4 ±463.35 | 678.6 | 1165.0 ±636.23 | 1046.0 | <0.001 |
| | LoEAP [mm] | 32.5±11.73 | 30.6 | 35.5±12.04 | 35.0 | 0.049 |
| | LoEML [mm] | 30.6±14.77 | 28.2 | 40.7±16.31 | 38.2 | <0.001 |
| 0 (n = 262) | SP [mm] | 570.6 ±177.45 | 549.7 | 644.3±197.69 | 630.1 | 0.002 |
| | V [mm/s] | 19.8±6.15 | 19.1 | 22.5±7.09 | 22.0 | 0.001 |
| | AoE [mm²] | 886.9 ±533.30 | 758.5 | 1029.7 ±555.79 | 947.9 | 0.021 |
| | LoEAP [mm] | 32.6±11.34 | 31.2 | 35.2±12.58 | 33.1 | 0.11 |
| | LoEML [mm] | 33.8±14.42 | 31.7 | 36.8±15.35 | 35.2 | 0.16 |
| -1 (n = 10) | SP [mm] | 430.4±81.81 | 403.2 | 612.2±225.63 | 578.2 | No test was performed due to the small size of the group |
| | V [mm/s] | 15.0±2.87 | 14.0 | 21.3±7.86 | 20.2 | |
| | AoE [mm²] | 652.6 ±275.18 | 651.5 | 679.7±345.72 | 687.6 | |
| | LoEAP [mm] | 27.1±11.10 | 24.9 | 28.4±12.22 | 25.9 | |
| | LoEML [mm] | 30.4±3.75 | 31.4 | 29.9±9.47 | 29.1 | |

Note: n—number of children; SD–standard deviation; Me–median; SP–sway path length of COP; V–average velocity of COP; AoE–area of the ellipse; LoEAP–length of the ellipse in the anterior-posterior direction; LoEML–length of the ellipse in the medial-lateral direction; (1) tall-statured children; (0) normal-statured children; (-1) low-statured children. Statistical significance was set at p<0.05.

the groups of low-, normal- and tall-statured children (Table 2). Girls achieved lower results of these parameters than boys regardless of their body height. For AoE in the groups of normal- and tall-statured children, Mann-Whitney U test showed a statistically significant sex effect (U = 7157.5, p = 0.021; U = 2091, p<0.001; respectively). A statistically significant effect of sex was observed for LoEAP and LoEML in the group of tall-statured children (U = 2657.5, p = 0.049; U = 1957, p<0.001; respectively). In the case of LoEAP and LoEML in the group of normal-statured children, sex effect was not significant (p = 0.11; p = 0.16, respectively).

## Discussion

The aim of this study was to assess differences in postural stability depending on sex of 5-year-old children with different body heights. This aim was achieved through recording the amount of sways within 30 seconds in the same age group.

The present study revealed that sex-related differences were found during a natural stance for COP shift parameters (sway path length of COP and average velocity of COP) and COP surface area parameters (area of the ellipse, length of the ellipse in the anterior-posterior direction and length of the ellipse in the medial-lateral direction). Girls achieved lower values of all COP parameters than boys. It means that girls maintained a two-legged standing position with lower sway velocity, a smaller range of sway and a smaller area of the ellipse than their male counterparts.

The findings of the present study are in line with the studies which show that girls at different ages have lower values of COP parameters. Geldhof et al. [29] noted that girls aged 9–10

achieved lower values of sway velocity compared to boys, which indicates better postural control in girls. Lee and Lin [30] revealed that boys had significantly larger mean radius of COP distributions than girls in the eyes-open and eyes-closed conditions. With respect to sex differences, boys had significantly poorer single-leg stance postural stability than girls. Smith et al. [31] showed that girls aged 8–12 had better postural stability than boys in normal standing conditions (hard surface, eyes open and looking straight ahead). Differences between sexes are reflected in lower path velocity, smaller radial displacement and lower area velocity of COP in girls than in boys [31].

The results of this study show that girls had better postural stability than boys. In line with this, Steindl et al. [32] also noted that girls were able to maintain balance more accurately than boys in the age groups below 11–12 years, with the exception of the group of 5-6-year-olds. Females showed a greater rate of improvement in stability until 11–12 years of age [32]. Venetsanou and Kambas [22] reported that girls in preschool age outperformed boys in the following tests: standing on the preferred leg on the floor, standing on the preferred leg on a balance beam, standing on the preferred leg on a balance beam–eyes closed, walking forward heel-to-toe on a walking line, walking forward heel-to-toe on a balance beam. In turn, boys had significantly higher scores in walking forward on a balance beam. However, according to the authors, the effect of sex on BOTMP (Bruininks-Oseretsky Test of Motor Proficiency) balance items was weak, indicating that the observed superiority of the girls was not of great importance [22]. Paniccia et al. [33] also found sex-based differences in postural stability performance between girls and boys in athletes aged 9–12, whereby girls had better postural stability compared to boys in conditions in which visual information was presented.

Differences between sexes are equivocal in the literature. Other researchers report no significant sex differences in balance skills at the age of 5–6 [16, 32, 34, 35]. It may be assumed that differences in the obtained results may depend on the use of different methods, measuring platforms or tests, different numbers of study participants or different age groups.

In the 5$^{th}$ year of life, an accelerated increase in body length (changes in the somatic development) which makes the body slimmer is observed. Significant differences in physique (body dimensions) are visible, which creates different conditions for maintaining a stable body posture. In order to minimise differences in body height, body height values of study participants were classified into three categories determined by the WHO [28]. The present study showed that significant effects of sex on postural sway parameters were noted in the groups of normal- and tall-statured children.

The present study suggests that sex differences in postural stability in children with similar body height are not influenced by their physique. Sex differences in postural stability among children may explain maturational differences of central nervous structures [16, 36]. Girls seem more capable of integrating their sensory inputs under normal standing conditions, while boys treat each sensory input somewhat separately with less integration [37]. Moreover, Hirabayashi and Iwasaki [38] showed that girls aged 7–8 were significantly superior to boys of the same age in the use of the vestibular cues under the condition of no visual cues and inaccurate somatosensory input. In line with this, Smith et al. [31] and Peterson et al. [37] also suggested that girls had better postural control under normal standing conditions (with information obtained by the vestibular system). Females up to the age of 11–12 developed sensory systems earlier than males [32]. Girls are better at skills which require balance and rhythm, which are stimulated by the vestibular system. In turn, boys tend to spend more time engaged in moderate-to-vigorous physical activity and achieve better results in activities which require more speed and strength [21, 22, 23, 24]. Consequently, boys may be at an advantage over girls in terms of proactive balance as they can compensate worse balance with greater muscle strength [35]. It is possible that maturational slowness of the vestibular function seen

in young boys is one of the factors responsible for the fact that boys are prone to be more active than girls [32, 38].

## Limitations

A limitation of the present study is that the analysis of differences in postural stability was limited to two factors, i.e. sex and height, in one age group. However, these limitations did not affect the value of the presented results significantly. Further studies require consideration of other variables.

## Study strengths

Studies investigating preschoolers' postural stability or balance skills do not provide a clear picture of children's postural stability at the age of 5. The data from this study provide evidence that postural stability differs between sexes already at this stage of development. Due to a large number of the study participants, the obtained results may have a normative value for children in this age group and be the basis for comparative analysis in different populations of children. Lastly, height status was classified according to age and sex into low, normal- and tall-statured, which enabled us to compare postural stability between girls and boys taking into account similar height.

## Conclusion

The current study presents sex differences in postural stability of 5-year-old children. Sex-related differences were found during a natural stance for COP shifts and COP surface area parameters. Girls maintained a two-legged standing position with lower sway velocity, with a smaller range of sway and a smaller area of the ellipse than their male counterparts. Normal- and tall-statured girls demonstrated better postural stability significantly more often than boys.

## Supporting information

**S1 Fig. Boxplot graphics—Values of COP parameters among children with regard to sex and significance levels from the Mann-Whitney U test.**
(TIF)

**S1 File. Dataset.**
(XLS)

## Author Contributions

**Conceptualization:** Magdalena Plandowska, Małgorzata Lichota, Krystyna Górniak.

**Data curation:** Magdalena Plandowska.

**Formal analysis:** Magdalena Plandowska.

**Investigation:** Magdalena Plandowska, Krystyna Górniak.

**Methodology:** Magdalena Plandowska, Małgorzata Lichota, Krystyna Górniak.

**Project administration:** Krystyna Górniak.

**Resources:** Magdalena Plandowska, Małgorzata Lichota, Krystyna Górniak.

**Visualization:** Magdalena Plandowska.

**Writing – original draft:** Magdalena Plandowska.

**Writing – review & editing:** Magdalena Plandowska, Małgorzata Lichota, Krystyna Górniak.

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
