## [Decision Letter · Decision Letter 0]

17 Sep 2019

PONE-D-19-20547

Postural stability of 5-year-old children with different levels of body height

PLOS ONE

Dear Mrs Plandowska,

Thank you for submitting your manuscript to PLOS ONE. After careful consideration, we feel that it has merit but does not fully meet PLOS ONE’s publication criteria as it currently stands. Therefore, we invite you to submit a revised version of the manuscript that addresses the points raised during the review process.

Both Reviewers identified serious problems with the submitted manuscript. After reading the reviews and the manuscript, I am in agreement with the Reviewers. Reviewer 2 recommends that your manuscript be rejected. This seems to me a reasonable recommendation, yet I am inclined to offer you the opportunity to revise and resubmit. Please take the reviews very seriously. It seems to me that a successful revision is possible, but will require very substantial revision of many aspects of the manuscript, as well as a detailed Cover Letter addressing the specific issues raised by the Reviewers. 

We would appreciate receiving your revised manuscript by Nov 01 2019 11:59PM. To enhance the reproducibility of your results, we recommend that if applicable you deposit your laboratory protocols in protocols.io, where a protocol can be assigned its own identifier (DOI) such that it can be cited independently in the future. For instructions see: http://journals.plos.org/plosone/s/submission-guidelines#loc-laboratory-protocols

We look forward to receiving your revised manuscript.

Kind regards,

Thomas A Stoffregen, PhD

Academic Editor

PLOS ONE

Journal Requirements:

1. We noticed you have some minor occurrence of overlapping text with the following previous publications, which needs to be addressed:

https://journals.plos.org/plosone/article?id=10.1371%2Fjournal.pone.0176556

https://www.sciencedirect.com/science/article/abs/pii/S0966636215009650

https://content.sciendo.com/view/journals/hukin/33/1/article-p25.xml

https://www.sciencedirect.com/science/article/abs/pii/0966636296828499

https://www.brainanddevelopment.com/article/0387-7604(95)00009-Z/pdf

In your revision ensure you cite all your sources (including your own works), and quote or rephrase any duplicated text outside the methods section. Further consideration is dependent on these concerns being addressed.

2. Data availability issue. In your statement you say "All relevant data are within the manuscript", but as we explain in http://journals.plos.org/plosone/s/data-availability#loc-faqs-for-data-policy you should provide the individual data points behind means, medians and variance measures presented in the results, tables and figures, and not just those summary statistics. Please provide these underlying participant-level data in a supporting information file or public repository, taking care not to include identifying information (see http://www.bmj.com/content/340/bmj.c181.long); if these data cannot be publicly deposited or included in the supporting information, e.g. due to patient privacy, legal reasons, or being provided by a third party, please explain why and explain how researchers may access them. Note that authors should not be the sole named individuals responsible for ensuring data access.

Reviewers' comments:

Reviewer's Responses to Questions

**Comments to the Author**

1. Is the manuscript technically sound, and do the data support the conclusions?

Reviewer #1: Partly

Reviewer #2: No

2. Has the statistical analysis been performed appropriately and rigorously? 

Reviewer #1: I Don't Know

Reviewer #2: Yes

3. Have the authors made all data underlying the findings in their manuscript fully available?

Reviewer #1: Yes

Reviewer #2: No

4. Is the manuscript presented in an intelligible fashion and written in standard English?

Reviewer #1: Yes

Reviewer #2: Yes

5. Review Comments to the Author

Reviewer #1: Dear authors, I enjoyed reading your paper. The large sample of 5 year olds can provide a good insight into postural control at that age. Differences between genders are equivocal in the literature and therefore studies are needed. However, I do have some concerns about your paper. The paper is not clear from the linguistic standpoint. I think, overall it would be good to have the document proof-read. The introduction includes an excessive amount of information about the COP that experts do not need while other parts of your paper are lacking information. In terms of the methodology, I have reservations regarding the use of three ranges of COP velocity since those could be chosen randomly as well as parts of the statistical analysis. The discussion seems weak in terms of the amount of information coming from your study. In general, the paper lacks of justifications and citations. A major review is needed. You can find a more detailed review below.

Major issues

1. Lines 80-105. There is over one page regarding basic information about the COP and how to obtain it. Experts in the field do not need that information. You should reduce this part.

2. Lines 94-104. You are using very old references. The studies are relevant in our field but, there is more recent literature better explaining your points and correcting some of the affirmations from those authors. For example, I suggest these two articles:

a. Motor mechanisms of balance during quiet standing, by David A. Winter et al (same author that you cite but more recent).

b. Human Postural Control, by Yury Ivanenko and Victor S. Gurfinkel

3. Lines 137-139. You are using the term “levels” for postural stability and for body height. I would rather suggest using greater/worse/more/less/higher/lower instead of levels since there are no established levels of either.

4. Lines 146-147. “parents written consent” is not usually listed as inclusion criteria. Instead, you can add a separated sentence like the following “Informed consent was obtained from the parents/guardian of all the children enrolled in the study.” This goes along the Stage I – preparation section

5. There are several language mistakes in the paper. You should have it corrected by a language expert.

6. Lines 198-199. Can you justify the reason why you use the variables of height and width of the ellipse? Instead of those two, in order to analyse the COP in the anterio-posterior and medio-lateral direction, is there a reason why you did no use COP velocity?

7. Lines 202 -218. What you describe as a qualitative assessment is wrong. You have used quantitative measures (COP velocity) and chose three ranges to compare them. Have you based your ranges on any previously published paper? If not, what is the reasoning behind allocating certain ranges to low, average, high postural stability?

8. Line 235-243. Can you justify the statistical analysis used for your variables? Why did you use Chi-Square statistics?

9. Your discussion includes a lot of information from different papers and researchers but little from your own study. You are not addressing the results from each of your five parameters and how each of them are relevant to postural control.

Minor issues

1. Lines 120-121. By this sentence it looks like you say that adult levels of postural control are attained at 6 years of age. Studies have shown that adult levels of postural control are reached much later that 6 years old (check article below). At 6 years though, children do have mastered adult levels in some specific aspects of postural control. You should speficy more or rephrase.

Barozzi S, Socci M, Soi D, Di Berardino F, Fabio G, Forti S, et al. Reliability of postural control measures in children and young adolescents. Eur Arch Otorhinolaryngol. 2014 Jul;271(7):2069–77.

2. Line 136. Energy is not a physical ability, please rephrase.

3. Line 140. “mount of sway” not sways.

4. Line 164. In your Initial medical check-ups you assess general health as an exclusion criteria and that does not appear listed in the exclusion criteria previously mentioned. Please clarify whether this was or not another exclusion criteria.

5. Line 180-181. The sentence is confusing, please check language grammar.

6. Lines 188-199. You mention three sway parameters and then you list 5: COP area, ellipse area, COP path, COP velocity, height of ellipse, width of ellipse

7. Review the bibliography. There are some mistakes on numbers:

5: there is a coma after the title, it should be a point.

19: Remove “dos Santos Cardoso de” for the first author, it is repeated (Sa, C.), the title starts with “Development” and not “Developmental”.

24: There is a coma after the authors, it should be a point.

Reviewer #2: The authors employed standard methods and analysis to quantify the postural sway of 5-year-old boys and girls during a task where they tried, supposedly, to stay as still as possible for 30 s. The data collection and analysis seem to be correct in the sense they are not very different than the ones used in other studies in the area.

The main problem with the present study is that the authors try to use these measurements to quantify postural stability and use that to differentiate the degree of stability between girls and boys.

After the authors tried to interpret their results with more postural-control centered explanations, they laconically recognize that "The postural stability of girls is better because boys seem to be less attentive.".

So, I question the significance of the results if one can trivialize them by simply stating that they were obvious since a 5-year-old boy can't stand still. If the authors still want to study the stability of the postural control in children, they should at least provide an independent measure of attention.

At the current state, this study provides very limited novel information.

The authors declared that "all data are fully available without restriction", but I couldn't find any information about where to access the data.

6. PLOS authors have the option to publish the peer review history of their article (what does this mean?). If published, this will include your full peer review and any attached files.

Reviewer #1: No

Reviewer #2: No

---

## [Author Response · Author response to Decision Letter 0]

1 Nov 2019

We would like to thank for the review and comments regarding our manuscript titled “Postural stability of 5-year-old children with different levels of body height” written by Magdalena Plandowska, Małgorzata Lichota and Krystyna Górniak.

Thank you for a detailed analysis of our work and for all the comments which will undoubtedly lead to increasing the value of this article. 

The changes suggested by the Reviewers have been enlisted below. In order to facilitate the assessment of the changes made in the revised version of the manuscript, they have been highlighted in grey. We hope that the changes will be satisfactory for the Reviewers and will make the paper methodologically correct and interesting for the readers.

Response to Reviewer # 1

Major issues

1. Regarding the comment: There is over one page regarding basic information about the COP and how to obtain it. 

I agree that the Introduction section contains too much basic information about the COP. As suggested, this part has been reduced (lines 79-90).

2. Regarding the comment: You are using very old references. 

Thank you for information about publications. References have been changed.

3. Regarding the comment: You are using the term “levels” for postural stability and for body height. I would rather suggest using greater/worse/more/less/higher/lower instead of levels since there are no established levels of either.

Thank you for your comment regarding the term “levels” for postural stability and for body height. I agree that the there are no established levels of either. I have taken them into account. According to the reviewer’s suggestion, the term “levels” has been removed. The title has been reedited. Instead of the term “levels”, I decided to use better/worse postural stability.

4. Regarding the comment: “parents written consent” is not usually listed as inclusion criteria. 

Thank you for your comment. We have taken it into account. The Participants section has been reedited (lines 130-137).

5. Regarding the comment: There are several language mistakes in the paper. 

Thank you for your comment. As suggested, the article has been edited by a native speaker.

6. Regarding the comment: Can you justify the reason why you use the variables of height and width of the ellipse?

The reason is the limitations of the measuring platform. The equipment does not provide detailed values of COP velocity in anterior-posterior (AP) and medial-lateral (ML) directions. Therefore, in order to analyse the COP in the anterior-posterior and medial-lateral direction, I used length of ellipse in the anterior-posterior direction (LoEAP, height of the ellipse) and length of ellipse in the medial-lateral direction (LoEML, width of the ellipse). I think that the selected parameters properly describe shifts of the COP and the surface of the ellipse (lines 173-186).

7. Regarding the comment: What you describe as a qualitative assessment is wrong. Have you based your ranges on any previously published paper? If not, what is the reasoning behind allocating certain ranges to low, average, high postural stability?

Thank you for your comments. A big number of study participants (435) made it possible to determine the levels of postural stability on the basis of the available methods of descriptive statistics. I thought that the selected criteria of the level of postural stability may constitute a reference to the assessment of postural stability in children from this age group. But I have not based ranges on any previously published paper. I decided to remove a qualitative assessment. Therefore, the section Stance analysis (lines 173-186) and the section Results were reedited (lines 215-240).

8. Regarding the comment: Can you justify the statistical analysis used for your variables? Why did you use Chi-Square statistics?

The Shapiro-Wilk test was used to analyze whether the variables had a normal distribution. Data normality was rejected, so the Mann-Whitney U test was used to examine differences between girls and boys. 

The Chi square test was used to identify significant differences in the percentage of children with different postural stability. I decided to remove qualitative assessment, so the Chi square test was removed, too.

9. Regarding the comment: Your discussion includes a lot of information from different papers and researchers but little from your own study. You are not addressing the results from each of your five parameters and how each of them are relevant to postural control.

Thank you for your comments. The section Discussion has been reedited.

Minor issues

1. Regarding the comment: By this sentence it looks like you say that adult levels of postural control are attained at 6 years of age. You should speficy more or rephrase.

The reviewer`s suggestion has been taken into account in the manuscript. This sentence has been corrected (lines 105-106).

2. Regarding the comment: Energy is not a physical ability, please rephrase.

It has been corrected (line 120-121).

3. Regarding the comment: “mount of sway” not sways.

Thank you for your comment. It has been corrected according to the suggestion (line 125).

4. Regarding the comment: In your Initial medical check-ups you assess general health as an exclusion criteria and that does not appear listed in the exclusion criteria previously mentioned. Please clarify whether this was or not another exclusion criteria.

Thank you for this comment. Bad general health was another exclusion criterion (lines 132-134).

5. Regarding the comment: The sentence is confusing, please check language grammar.

Thank you for this comment. This sentence has been reedited (166-168).

6. Regarding the comment: You mention three sway parameters and then you list 5: COP area, ellipse area, COP path, COP velocity, height of ellipse, width of ellipse.

Thank you for this comment. A technical error.

7. Regarding the comment: Review the bibliography. There are some mistakes.

Thank you for your comment. The bibliography has been corrected.

Response to Reviewer #2: 

Thank you for reading our article. According to the comments of the reviewers, the manuscript has been corrected.

---

## [Decision Letter · Decision Letter 1]

19 Nov 2019

PONE-D-19-20547R1

Postural stability of 5-year-old girls and boys with different body heights

PLOS ONE

Dear Mrs Plandowska,

Thank you for submitting your manuscript to PLOS ONE. After careful consideration, we feel that it has merit but does not fully meet PLOS ONE’s publication criteria as it currently stands. Therefore, we invite you to submit a revised version of the manuscript that addresses the points raised during the review process.

Reviewer 1 saw your original submission, and is satisfied with your revisions. Reviewer 2 is new. Reviewer 2 raises a number of minor issues that could benefit from clarification. Please make the requested changes, and I will be happy to accept your paper for publication in PLOS ONE.

We would appreciate receiving your revised manuscript by Jan 03 2020 11:59PM. To enhance the reproducibility of your results, we recommend that if applicable you deposit your laboratory protocols in protocols.io, where a protocol can be assigned its own identifier (DOI) such that it can be cited independently in the future. For instructions see: http://journals.plos.org/plosone/s/submission-guidelines#loc-laboratory-protocols

We look forward to receiving your revised manuscript.

Kind regards,

Thomas A Stoffregen, PhD

Academic Editor

PLOS ONE

Additional Editor Comments (if provided):

Reviewer 1 saw your original submission, and is satisfied with your revisions. Reviewer 2 is new. Reviewer 2 raises a number of minor issues that could benefit from clarification. Please make the requested changes, and I will be happy to accept your paper for publication in PLOS ONE.

Reviewers' comments:

Reviewer's Responses to Questions

**Comments to the Author**

1. If the authors have adequately addressed your comments raised in a previous round of review and you feel that this manuscript is now acceptable for publication, you may indicate that here to bypass the “Comments to the Author” section, enter your conflict of interest statement in the “Confidential to Editor” section, and submit your "Accept" recommendation.

Reviewer #1: All comments have been addressed

Reviewer #3: (No Response)

2. Is the manuscript technically sound, and do the data support the conclusions?

Reviewer #1: Yes

Reviewer #3: Yes

3. Has the statistical analysis been performed appropriately and rigorously? 

Reviewer #1: Yes

Reviewer #3: Yes

4. Have the authors made all data underlying the findings in their manuscript fully available?

Reviewer #1: Yes

Reviewer #3: Yes

5. Is the manuscript presented in an intelligible fashion and written in standard English?

Reviewer #1: Yes

Reviewer #3: Yes

6. Review Comments to the Author

Reviewer #1: After rereading the changes, I have found that you have addressed all the points that I mentioned on my first review. The large sample was already valuable and now it is well written and methodologically sound from my point of view.

Reviewer #3: Page 2, line 31

“Gender” is utilized throughout the paper, but the proper term for what was studied is “sex.”

Page 7, line 167

Other experiments of postural sway have utilized marked stance widths. Were children told to stand “normally”? Later (page 12, line 272) states that girls had a smaller stance, which indicates to me that the participants were not given a “set” width. A more detailed explanation of methods on page 7 is needed.

Page 9, line 197

The phrase “statistically significant” indicates that some form of analysis was used, but this doesn’t say what analysis. Mann-Whitney U is stated later, but proper reporting is (U = xxx, p=0.02). Clarity is needed here.

Page 10, line 220

This also needs to proper statistical reporting (U=xxx, p=.xxx). This is seen throughout the rest of the manuscript.

Page 14, line 312

This paragraph seems like it should be in the into, not the discussion.

Page 14, line 315

I am confused on why this paragraph opens up with “environment and socio-cultural factors,” and then presents no discussion of these factors.

What is discussed in this section seems like it would fit in the introduction as motivation.

A discussion of said environment and socio-cultural factors would be appropriate here, and would help provide insight into the differences being seen.

Overall, I found this paper to be a worthy contribution to the scientific community. My main issue is that the paper has several minor issues that can be revised without too much trouble. The most concerning of these is the discussion, which I feel contains information that should have been used to motivate the study in the introduction. This section limits the impact the article has by reflecting on the findings of others, instead of discussing the study's findings to their full extent.

7. PLOS authors have the option to publish the peer review history of their article (what does this mean?). If published, this will include your full peer review and any attached files.

Reviewer #1: Yes: Roberto Izquierdo-Herrera

Reviewer #3: No

---

## [Author Response · Author response to Decision Letter 1]

7 Dec 2019

Thomas A Stoffregen, PhD

Academic Editor

PLOS ONE

We would like to thank for the review and comments regarding our manuscript titled “Postural stability of 5-year-old girls and boys with different body heights” written by Magdalena Plandowska, Małgorzata Lichota and Krystyna Górniak.

Thank you for a detailed analysis of our work and for all the comments which will undoubtedly lead to increasing the value of this article. 

The changes suggested by the Reviewer 3 have been enlisted below. In order to facilitate the assessment of the changes made in the revised version of the manuscript, they have been highlighted in grey. We hope that the changes will be satisfactory for the Reviewer and will make the paper methodologically correct and interesting for the readers.

Response to Reviewer # 1

Thank you for a detailed analysis of our work and for all the comments which undoubtedly leaded to increasing the value of this article. 

Response to Reviewer # 3

1. Regarding the comment:“Gender” is utilized throughout the paper, but the proper term for what was studied is “sex.”

Thank you for your comment. We have taken it into account.

2. Regarding the comment: Page 7, line 167 - Other experiments of postural sway have utilized marked stance widths. Were children told to stand “normally”? Later (page 12, line 272) states that girls had a smaller stance…. 

Thank you for your comment. The sentence (page 12,line 279-281) has been corrected.

3. Regard the comment: Page 9, line 197 - The phrase “statistically significant” indicates that some form of analysis was used, but this doesn’t say what analysis. Mann-Whitney U is stated later, but proper reporting is (U = xxx, p=0.02). 

Thank you for your comment. It has been corrected according to the suggestion.

4. Regard the comment: Page 10, line 220 - This also needs to proper statistical reporting (U=xxx, p=.xxx). This is seen throughout the rest of the manuscript.

Thank you for your comment. We have taken it into account.

5. Regard the comment: Page 14, line 312 - This paragraph seems like it should be in the into, not the discussion.

Thank you for your comment. This paragraph has been included in the Introduction (page 5-6, line 118-121).

6. Regard the comment: Page 14, line 315 - I am confused on why this paragraph opens up with “environment and socio-cultural factors,” and then presents no discussion of these factors. What is discussed in this section seems like it would fit in the introduction as motivation. A discussion of said environment and socio-cultural factors would be appropriate here, and would help provide insight into the differences being seen.

Thank you for your comment. As suggested, this paragraph has been reduced (page 14, line 317-334) and part of it has been included in the Introduction (page 6, line 116-127). 

Yours sincerely,

Magdalena Plandowska

Jozef Pilsudski University of Physical Education in Warsaw,

Faculty of Physical Education and Health, Biala Podlaska, Poland

---

## [Editor Report · Decision Letter 2]

13 Dec 2019

Postural stability of 5-year-old girls and boys with different body heights

PONE-D-19-20547R2

Dear Dr. Plandowska,

We are pleased to inform you that your manuscript has been judged scientifically suitable for publication and will be formally accepted for publication once it complies with all outstanding technical requirements.

With kind regards,

Thomas A Stoffregen, PhD

Academic Editor

PLOS ONE
---

## [Editor Report · Acceptance letter]

19 Dec 2019

PONE-D-19-20547R2 

Postural stability of 5-year-old girls and boys with different body heights 

Dear Dr. Plandowska:

I am pleased to inform you that your manuscript has been deemed suitable for publication in PLOS ONE. Congratulations! Your manuscript is now with our production department. 

With kind regards,

on behalf of

Dr. Thomas A Stoffregen 

Academic Editor

PLOS ONE